# WNK2 may promote ovarian cancer progression by upregulating POU5F1B

**Fengjie Li**[1], **Yongqin Jia**[1], **Xiaoli Min**[1], **Pangyang Zhang**[1], **Yudi Li**[1], **Deng Li**[1], **Lanqin Cao**[2], **Yanzhou Wang**[1]*, **Zhiqing Liang**[1]*

1 Department of Gynecology and Obstetrics, The First Affiliated Hospital (Southwest Hospital) of Army Medical University, Chongqing, China, 2 Department of Obstetrics and Gynecology, Xiangya Hospital, Central South University, Changsha, China

* swog@tmmu.edu.cn (ZL); wyz@tmmu.edu.cn (YW)

## Abstract

Ovarian cancer (OC) remains the most lethal gynecologic malignancy. Our previous work showed that WNK lysine-deficient protein kinase 2 (WNK2) promotes OC cell proliferation and migration. To clarify the molecular basis of WNK2-driven OC progression, here, we performed transcriptome sequencing to identify WNK2-regulated mRNAs and noncoding RNAs. We validated candidate targets using qRT-PCR and Western blot analyses. Functional assays, including CCK-8, colony formation, and Transwell assays, evaluated the role of POU5F1B and its capacity to rescue the effects of WNK2 knockdown. POU5F1B is a promising OC therapeutic target, mediating WNK2-driven oncogenesis in xenograft models (n = 10). Because AKT acts downstream of POU5F1B, we examined AKT phosphorylation and found that POU5F1B displayed clear oncogenic activity in OC cells. WNK2 upregulated POU5F1B mRNA and protein levels, while POU5F1B overexpression reversed the tumor-suppressive effects caused by WNK2 depletion. Mechanistically, WNK2 silencing decreased AKT phosphorylation, which POU5F1B overexpression restored. Together, these results demonstrate that WNK2 promotes OC progression by upregulating the validated oncogene POU5F1B and activating AKT signaling. These findings establish WNK2 as an oncogenic driver and a promising therapeutic target in OC.

## 1. Introduction

Ovarian cancer (OC) remains the most lethal malignancy of the female reproductive system. In the United States, 20,890 new OC cases were diagnosed in 2020, leading to 12,730 deaths [1]. High-grade serous ovarian carcinoma (HGSOC), the most aggressive subtype, accounts for 70–80% of OC cases and causes most OC-related deaths [2]. Notably, the tumor marker CA-125 for OC suffers from low specificity and insufficient sensitivity, while imaging studies also exhibit a high rate of misdiagnosis. Consequently, approximately 70–75% of OC patients are initially diagnosed at

**Data availability statement:** All relevant data are within the manuscript and its Supporting information files.

**Funding:** The author(s) received no specific funding for this work.

**Competing interests:** The authors have declared that no competing interests exist.

an advanced stage (Stage III or IV) [3]. The initial treatment paradigm, centered on cytoreductive surgery and platinum-based chemotherapy, has been expanded by the introduction of novel agents, several new therapeutic approaches have emerged in recent years. These include antiangiogenic agents such as VEGF inhibitors, tyrosine kinase inhibitors, and folate receptor-targeted therapies [4]. In particular, PARP inhibitors have transformed the management of patients with BRCA1/2 mutations [5]. Despite these therapeutic advances, their overall impact has been constrained by significant challenges. Drug resistance, dose-limiting toxicity, and profound tumor heterogeneity frequently undermine clinical efficacy [6]. As a result, the 5-year survival rate for advanced OC has shown little improvement over the past decade, emphasizing the urgent need for novel molecular targets to develop more effective therapies.

WNK kinases, a family of serine/threonine kinases first identified in 2000, characterized by the absence of a conserved lysine residue in subdomain II, are increasingly implicated in cancer pathogenesis [7]. Most notably among them, WNK2 has been widely characterized as a tumor suppressor in various malignancies, including glioma, gastric, and pancreatic cancers, where it exerts its effects through mechanisms such as inhibiting Rac1 or ERK signaling [8–10]. In meningiomas, reduced WNK2 expression caused by abnormal CpG island methylation suppresses tumorigenesis [8]. However, we have uncovered a contrasting and provocative finding: WNK2 exhibits oncogenic activity in OC. We have provided evidence of high WNK2 expression not only in cell lines but also in tissue specimens. And high WNK2 expression was significantly positively correlated with the severity of tumor malignancy. Moreover, our functional assays revealed that WNK2 drives OC proliferation and metastasis in both cellular models and live animals [11]. The molecular basis for this dichotomy, wherein WNK2 acts as an oncogene in OC but as a tumor suppressor in other cancers, remains unclear. We conducted phosphorylation sequencing previously, and found WNK2 phosphorylates and thereby activates a cascade of downstream effectors, including transcription factors such as Jun, YAP1, STAT1, ATF1 and other kinases. To systematically unravel the downstream gene network orchestrated by this kinase and identify the specific genes it alters. Here, we performed RNA-seq analysis to achieve a deeper understanding of its functional mechanisms in OC. This study provides a more comprehensive evidence to establish WNK2 as a potent diagnostic and therapeutic target for OC.

## 2. Materials and methods

### 2.1 Cell culture

Human OC cell lines CAOV3 and A2780 were purchased from the American Type Culture Collection (ATCC, Manassas, VA, USA). Cells were maintained in Dulbecco's Modified Eagle's Medium (DMEM) supplemented with 10% fetal bovine serum (FBS) and 1% penicillin–streptomycin (Gibco, Gaithersburg, MD, USA) at 37 °C in a humidified incubator with 5% $CO_2$.

### 2.2 Cell transfection and lentivirus transduction

Empty vector and POU5F1B-overexpression (POU5F1B-OE) plasmids were synthesized by Tsingke Biotechnology Co., Ltd. (Beijing, China). The WNK2-overexpression

(WNK2-OE) plasmid and its matched empty vector were constructed by Genechem Co., Ltd. (Shanghai, China). Small interfering RNAs (siRNAs) targeting WNK2 and POU5F1B were purchased from RiboBio (Guangzhou, China) with the following sequences:

si-WNK2–1: CAA GGA CAA TGG AGC CATA; si-WNK2–2: GGA GTA TGC TAG GCT ATGA; and si-WNK2–3: CGA TGA AAT TGC CAC GTAT. si-POU5F1B-1: AGA AGT CCC AGG ACA TCAA, si-POU5F1B-2: CAC TGC AGA TCA GCC ACAT, and si-POU5F1B-3: CCC AGT CTC CGT CAT CACT. Transient transfection: Cells were transfected 24 hours after seeding. For each well, ~3 µg plasmid DNA or 50 nM siRNA was complexed with 5 µL Lipofectamine 3000 in serum-free medium according to the manufacturer's instructions. Six hours after adding the transfection mix, we replaced it with complete DMEM. Stable cell line generation: To generate stable WNK2 knockdown cells, we transduced CAOV3 cells with sh-WNK2 lentiviral particles. We then transduced those sh-WNK2 cells with POU5F1B-OE lentivirus to produce the following stable lines: sh-WNK2 + Vector and sh-WNK2 + POU5F1B-OE. Stable cells were selected with 2 µg/mL puromycin [12].

## 2.3 Transcriptome sequencing analysis

After WNK2 knockdown by siRNA transfection, we isolated total RNA using TRIzol™ reagent (Takara, Japan) from three biological replicates per group. The Beijing Novogene Institute (Beijing, China) conducted differential expression analysis using the DESeq2 R package (v1.20.0). Adjusted p-values (padj) were calculated with the Benjamini–Hochberg method, and genes with padj < 0.05 were defined as differentially expressed genes (DEGs) [13].

## 2.4 Quantitative real-time PCR

We extracted total RNA using TRIzol reagent and synthesized complementary DNA (cDNA) with a PrimeScript™ RT reagent kit (Takara, Japan). We quantified the mRNA levels of POU5F1B and WNK2 using quantitative real-time PCR (qRT-PCR) with TB Green Premix Ex Taq™ (Takara, Japan). GAPDH served as the internal control for normalization. The primer sequences were as follows: WNK2, F: 5'-TGG TTC ATC ATC TGT CCG-3'and R: 5'-AAG CTG GGT TGT TCC TT-3'. POU5F1B F: 5'-GAA CCG AGT GAG AGG CAA CC-3' and R: 5'-GAT GTG GCT GAT CTG CAG TGT-3'; GAPDH F: 5'-AGC CAC ATC GCT CAG ACAC-3' and R: 5'-TTA AAA GCA GCC CTG GTG AC-3'.

## 2.5 Western blotting

Cell lysates were prepared using radio-immunoprecipitation assay buffer (RIPA, Beyotime, Shanghai) supplemented with 1% protease inhibitor cocktail (Beyotime, Shanghai). Protein concentrations were measured with a bicinchoninic acid protein assay kit (Beyotime, Shanghai). Each sample was adjusted to equal protein amounts and volumes before electrophoresis. Proteins were then transferred onto polyvinylidene fluoride (PVDF) membranes (Thermofisher) at 250mA. After blocking nonspecific binding, we incubated the membranes with primary antibodies at 4 °C. The following antibodies were used: anti-WNK2 (cat: ab192397, 1:500, Abcam), anti-POU5F1B (cat: ab230429, 1:1,000, Abcam), anti-actin (cat: #3700, 1:1,000, CST), anti-mouse (cat: #7076, 1:5,000, CST), and anti-rabbit (cat: #7074, 1:5,000, CST).

## 2.6 Immunohistochemistry

We evaluated POU5F1B expression in 70 OC tissues and 10 adjacent non-tumor tissues using a commercial tissue microarray (HOvaC070PT01, Outdo Biotech). Immunohistochemistry (IHC) was performed with a commercial IHC kit (Gene Tech Company Limited) [14]. After deparaffinization at 60 °C for 30 min, we retrieved the antigen using citrate buffer and blocked nonspecific binding. Sections were incubated overnight at 4 °C with anti-POU5F1B antibody (cat: ab230429, 1:1,000; Abcam). After incubation with the secondary antibody, the sections were visualized using 3,3'-diaminobenzidine. The study was conducted in accordance with the Declaration of Helsinki principles and approved by the Research Ethics Committee of the Shanghai Outdo Biotech Company.

## 2.7 Proliferation and transwell assays

The cell viability was measured with a Cell Counting Kit-8 (CCK-8) assay kit and a colony formation assay. For the CCK-8 assay, 5000 transfected cells were seeded in 96-well plates, and viability was measured following the manufacturer's protocol. For the colony formation assay, we plated 2,000 cells per well in 12-well plates and incubated them for two weeks. Colonies were fixed and stained with crystal violet, and the number of colonies were then counted.

Transwell assays were analyzed by chambers with 8 µm pores to evaluate cell migration and invasion. A total of $2 \times 10^4$ cells suspended in serum-free medium were seeded in the upper chamber, while medium containing 10% serum was added to the lower chamber. After 48 h of incubation, we removed non-migrated cells from the upper surface by a cotton swab, then fixed the migrated cells with 4% paraformaldehyde, and stained them with crystal violet. Rescue experiments were conducted to determine whether POU5F1B could reverse the effects of WNK2 knockdown on cancer cell viability and invasion.

## 2.8 Nude mouse xenograft model

To examine whether WNK2 promotes OC malignancy through POU5F1B, we established a subcutaneous xenograft mouse model. Twenty female nude mice (six weeks old) were purchased from Huafukang Co., Ltd. (Beijing, China). All animal experiments were approved by the Institutional Animal Care and Use Committee of the Army Medical University (IACUC Approval No. KY2024087) and conducted at its Animal Center. Stable cell lines (sh-WNK2 + Vector or sh-WNK2 + POU5F1B-OE) were suspended in PBS ($1 \times 10^6$ cells/100 µL) and injected subcutaneously into the flanks of mice.

Tumor monitoring: Once palpable tumors formed, we measured tumor volumes twice weekly using calipers and calculated them as Volume = (Width² × Length)/ 2. Tumor growth curves were generated from these measurements. When tumor diameter reached 1 cm (approximately day 20), the mice were euthanized by cervical dislocation, and tumors were collected for analysis. All personnel involved in animal studies were trained in animal care and handling, and all procedures were performed under gas anesthesia. No mice died prior to the end of the experiment.

IHC: To assess POU5F1B expression in tumor tissues, we performed IHC analysis on the harvested xenografts.

## 2.9 Statistical analysis

We performed statistical analyses using GraphPad Prism 8.0. Data are presented as mean ± SD from three independent experiments. Differences between groups were analyzed using Student's *t*-test (paired or unpaired, as appropriate). $p < 0.05$ represents statistically significant.

## 3. Results

### 3.1 POU5F1B is the candidate target gene of WNK2

Our previous work identified a distinct role for WNK2 in OC compared with other malignancies. Analysis of TCGA data using the GEPIA portal showed that *WNK2* expression was lower in glioblastoma (GBM), Low-Grade Glioma (LGG), colon (COAD), pancreatic (PAAD), and rectal Adenocarcinoma (READ) (Fig 1A). To explore WNK2-dependent transcriptional regulation, we performed RNA sequencing on WNK2-knockdown OC cells. A heatmap illustrates the top 25 downregulated DEGs in si-WNK2 cells (Fig 1B). Among these, six DEGs - *POU5F1B*, *TRIB3*, *CDC42EP1*, *HES6*, *TYMP*, and *LARGE2* – were selected for further analysis based on their known roles in tumor progression (Fig 1C) [15–18]. qRT-PCR validation in *WNK2*-knockdown SKOV3 cells and *WNK2*-overexpressing A2780 cells confirmed POU5F1B as the most strongly regulated target of WNK2 (Fig 1D).

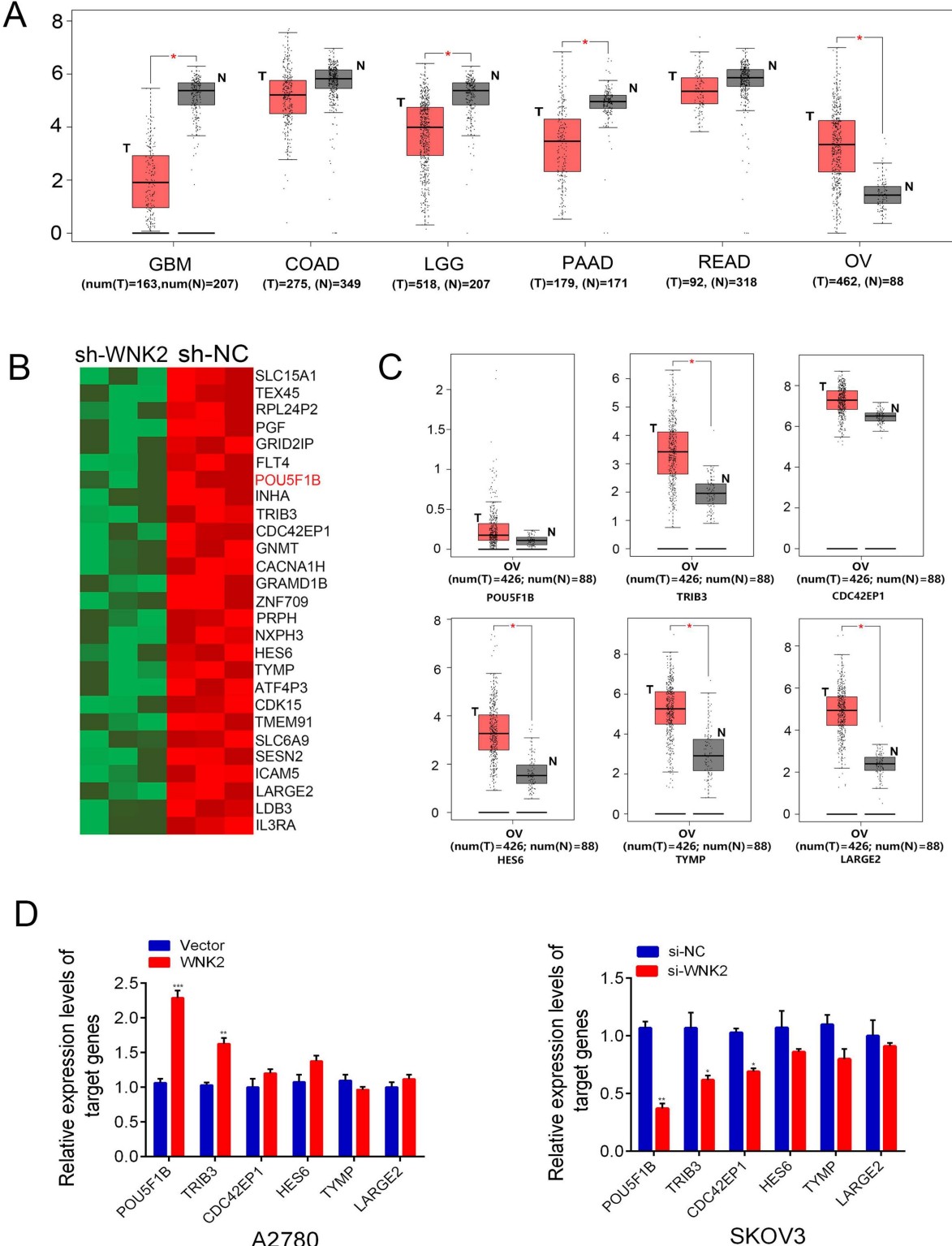

**Fig 1. POU5F1B is the potential target gene of WNK2.** A, The expression of WNK2 in glioblastoma, colon, pancreatic, colorectal adenocarcinoma and ovarian cancer. B, Heatmap of the DEGs. C, TCGA database shows the 6 selected DEGs are overexpressed in OC. D, RT-qPCR proved that mRNA level of POU5F1B significantly decreased or increased by WNK2 knockdown and overexpression.

## 3.2 WNK2 regulates POU5F1B positively at both mRNA and protein levels

RT-qPCR and western blot analyses demonstrated that WNK2 knockdown decreased POU5F1B expression, while WNK2 overexpression increased it (Fig 2A–2D). These findings indicate that WNK2 positively regulates POU5F1B at both the mRNA and protein levels. We next investigated the role of POU5F1B in OC progression.

## 3.3 POU5F1B functions as an oncogene in OC

Since public datasets lack POU5F1B protein expression data in OC, we evaluated its expression by immunohistochemistry (IHC). Representative staining showed low, moderate, and high POU5F1B expression patterns (Fig 3A). Analysis of the HOvaC070PT01 tissue microarray revealed significantly higher POU5F1B protein levels in OC tissues compared with matched adjacent normal tissues (Fig 3B–3D and Table 1).

We next examined the functional role of POU5F1B in OC using CCK-8, EdU, and colony formation assays. POU5F1B knockdown in A2780 and CAOV3 cells (Fig 4A) markedly reduced cancer cell proliferation (Fig 4B and 4C), whereas its overexpression promoted proliferation (Fig 5B and 5C). Transwell assays demonstrated that POU5F1B silencing inhibited, while its overexpression enhanced, cancer cell migration (Fig 4D and 5D).

## 3.4 WNK2 promotes OC growth and activates AKT through POU5F1B

To explore the interaction between WNK2 and POU5F1B, we established stable WNK2-knockdown (WNK2-KD) OC cell lines and transfected them with a POU5F1B overexpression plasmid. Immunoblotting confirmed efficient WNK2 silencing and restored POU5F1B expression in both A2780 and CAOV3 cells (Fig 6A and 6B). POU5F1B overexpression reversed the WNK2-KD-induced suppression of cell proliferation (CCK-8, colony formation; Fig 6C and 6D).

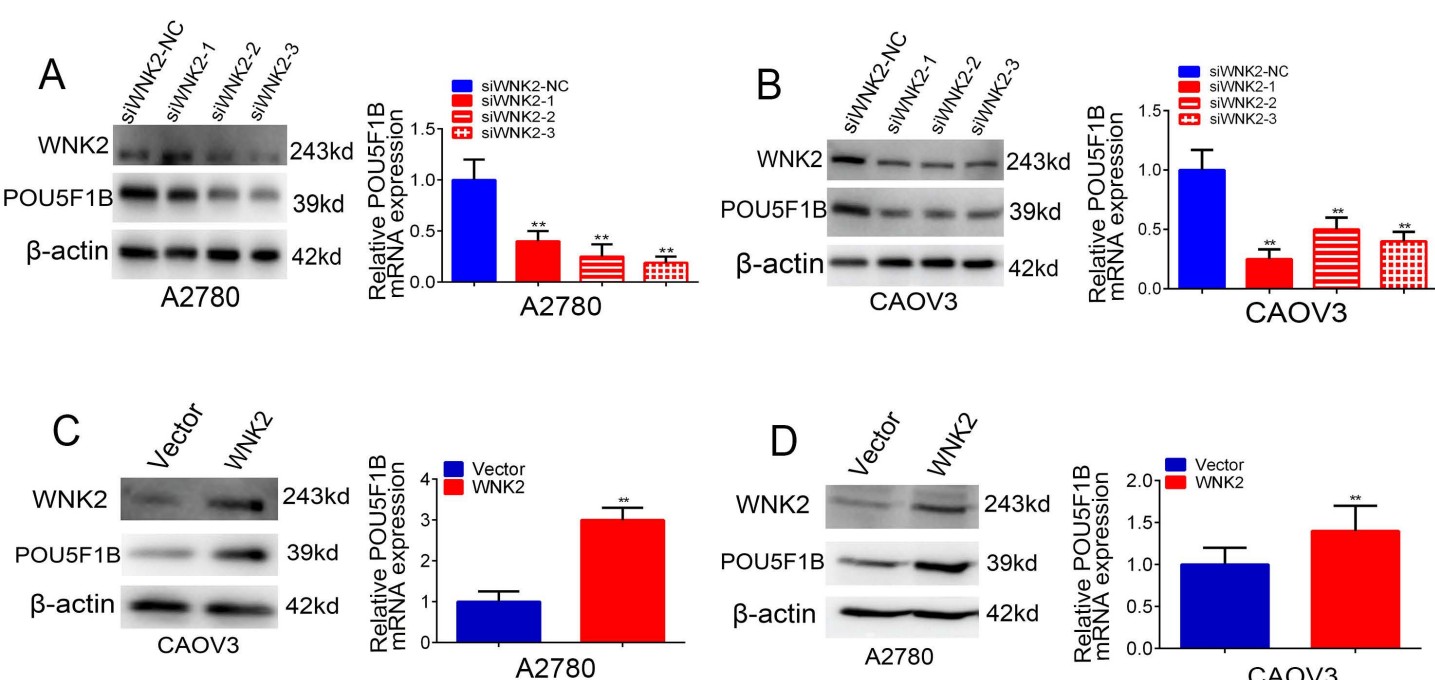

**Fig 2. POU5F1B is positively regulated by WNK2.** A-B, Western blot proved that protein expression of POU5F1B significantly decreased by WNK2 knockdown. C-D, WNK2 overexpression increased the protein expression of POU5F1B.

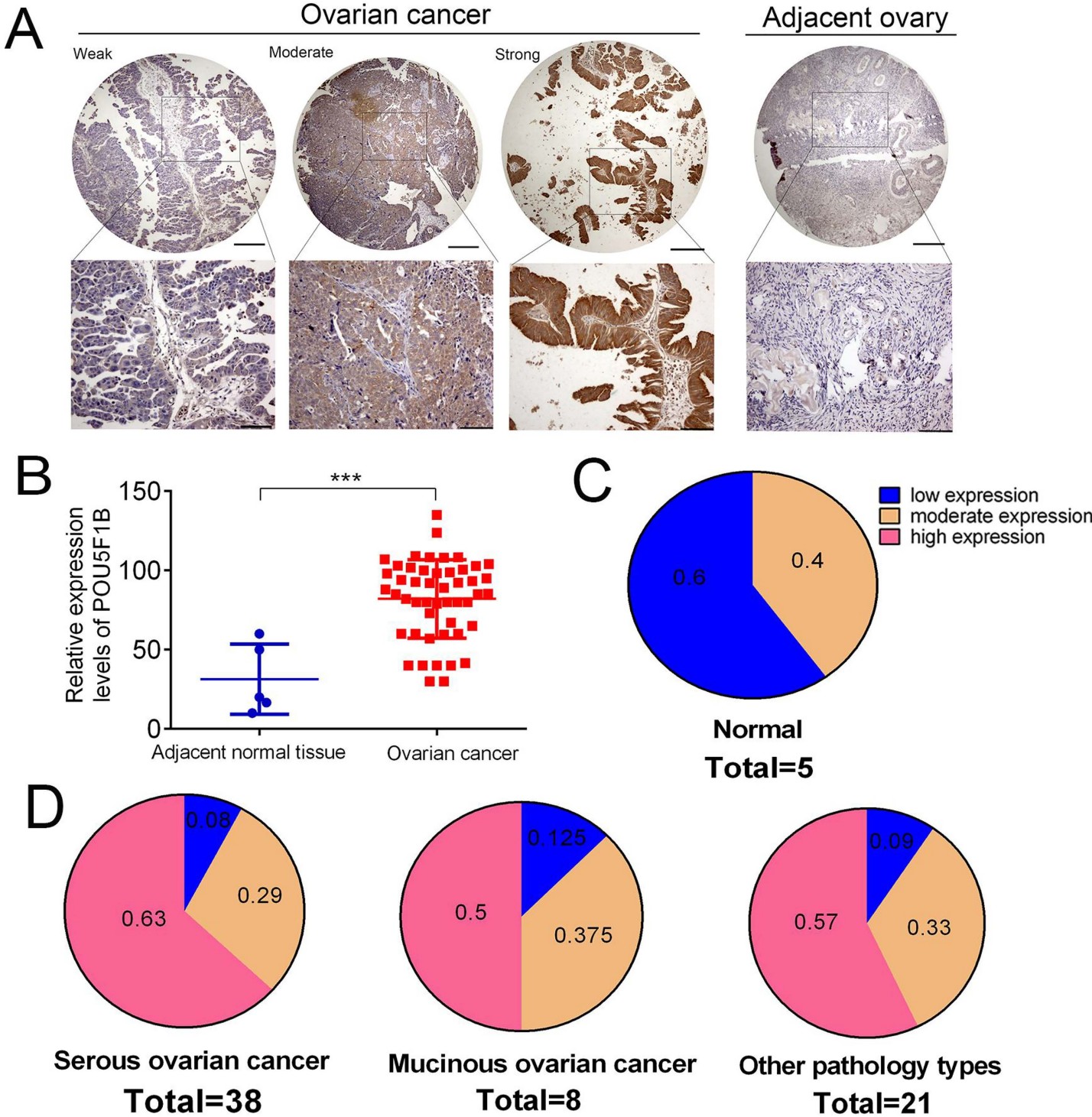

**Fig 3. POU5F1B is overexpressed in OC tissues.** A, The representative images of high, moderate and low expression of POU5F1B in OC and adjacent ovary tissues. B, The expression of POU5F1B is significantly overexpressed in OC tissues. C-D, The expression intensity proportion of POU5F1B in adjacent ovary and OC tissues.

**Table 1. POU5F1B is increased in OC tissues compared to adjacent ovary tissues.**

|  | High miR-324-3p expression | Low miR-324-3p expression | P-value |
|---|---|---|---|
| Pathology types |  |  |  |
| Boderline tumors | 0 | 5 |  |
| Serous carcinoma | 24 | 14 | 0.008* |
| Mucinous carcinoma | 4 | 4 | 0.068 |
| Other pathology types carcinoma | 12 | 9 | 0.024* |
| Ages (years) |  |  |  |
| ≧55 | 15 | 10 | 0.801 |
| <55 | 24 | 14 |  |

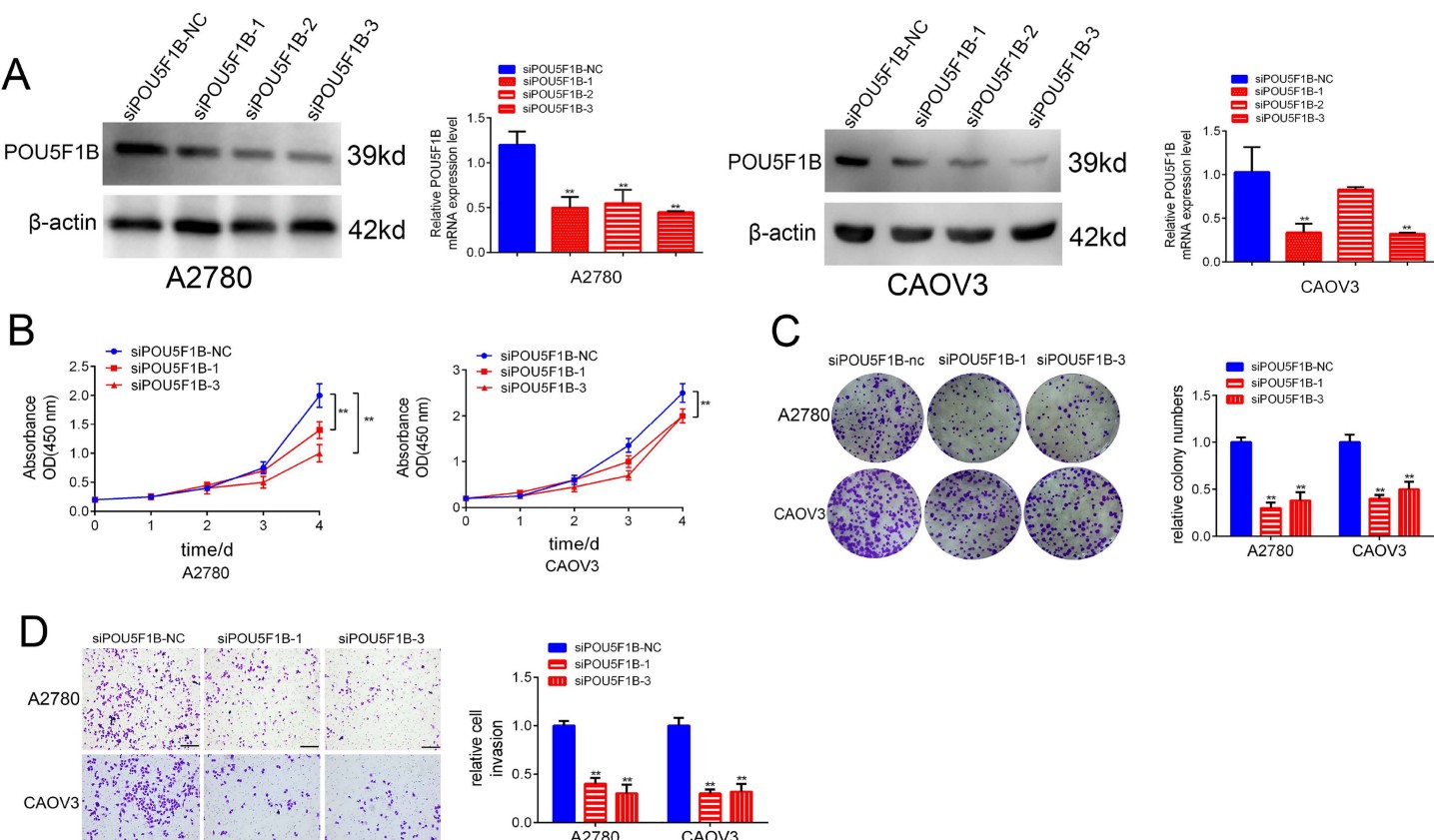

**Fig 4. POU5F1B knockdown suppresses the proliferation and invasion of OC cells.** A, Western blot confirmed the knockdown efficiency of POU5F1B in A2780 and CAOV3 cells. B, and C, CCK8 and colony formation show POU5F1B knockdown suppressed the proliferation of OC cells. D, Transwell assays show POU5F1B knockdown inhibits the migration of OC cells.

In xenograft models, POU5F1B re-expression restored tumor growth in WNK2-KD cells, as reflected by increased tumor volume and weight (Fig 6E–6G). IHC confirmed successful POU5F1B re-expression in tumors (Fig 6H). Since POU5F1B has been reported to activate the AKT pathway, we examined pathway activity. WNK2 knockdown decreased phospho-AKT (Ser473) levels without affecting total AKT, whereas POU5F1B overexpression rescued phospho-AKT suppression (Fig 6I).

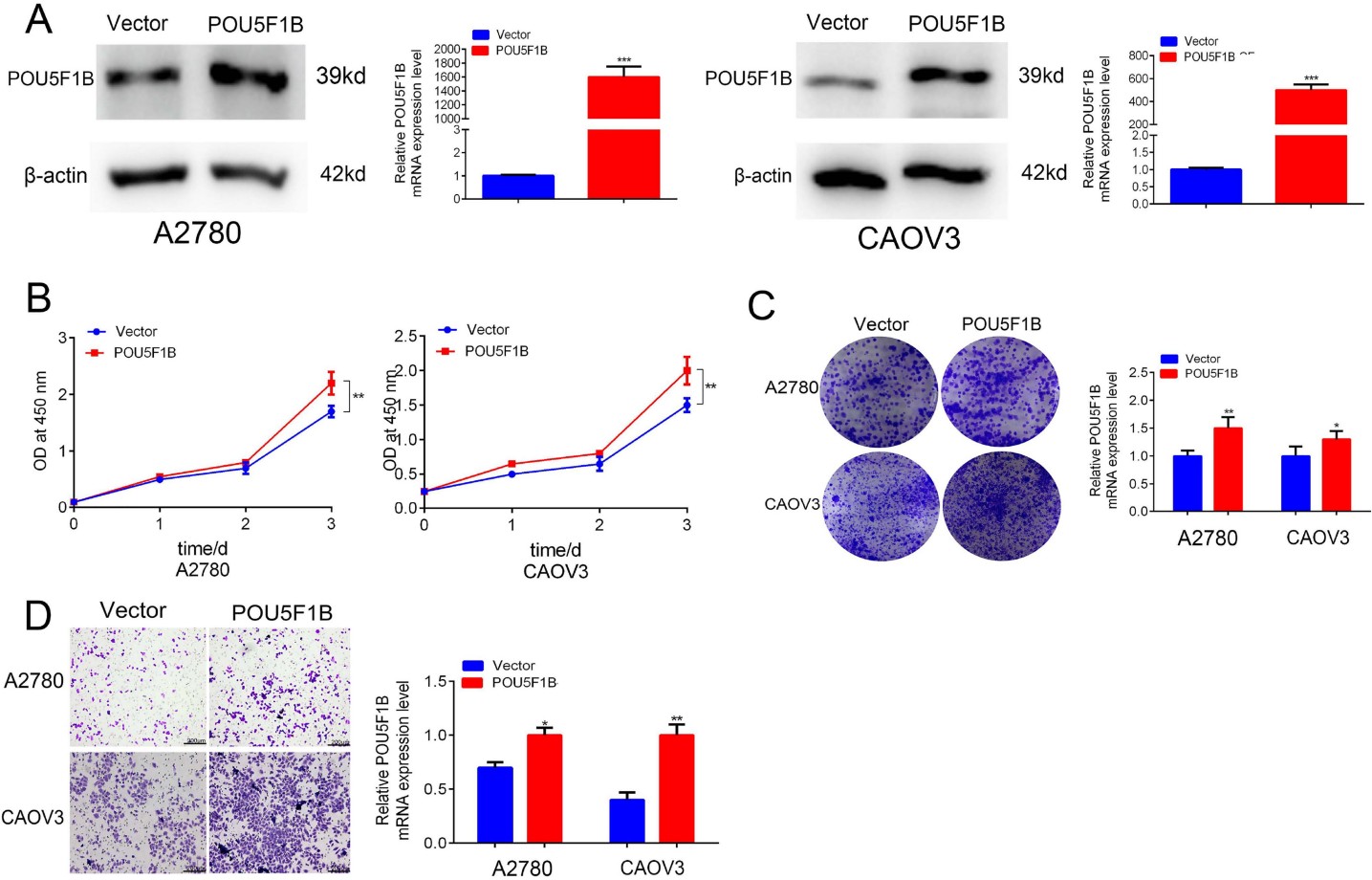

**Fig 5. Overexpression of POU5F1B promotes the proliferation and invasion of OC cells.** A, Western blot confirmed the overexpression of POU5F1B in A2780 and CAOV3. B and C, CCK8 and colony formation show POU5F1B overexpression promotes the proliferation of OC cells. D, Transwell assays show POU5F1B promotes the migration of OC cells.

## 4. Discussion

Cancer complexity often arises from the paradoxical functions of genes, where tumor suppressors may promote malignancy and oncogenes may exert suppressive effects [19]. WNK2 exemplifies this duality. Although identified as a tumor suppressor in glioblastoma, gastric, cervical, and pancreatic cancers [9]. WNK2 is markedly overexpressed in OC cells and patient tissues. Its overexpression correlates with higher tumor grade and poor patient prognosis. Functionally, WNK2 enhances OC cell proliferation and invasion in vitro and in vivo, yet the underlying molecular mechanisms remain unclear [11]. A deeper understanding of such context-dependent gene functions is critical for the development of targeted cancer therapies.

WNK2 encodes a cytoplasmic serine–threonine kinase characterized by coiled-coil domains and multiple PXXP and SH3 motifs, which suggest a multifaceted and context-specific biological role [20]. Primarily expressed in neurons, WNK2 regulates diverse physiological processes including circadian rhythm, electrolyte balance, cell survival, and proliferation [21,22]. To elucidate the molecular mechanism driving its oncogenic activity in OC, we employed high-throughput RNA sequencing (RNA-seq), which enables precise and comprehensive transcriptomic profiling [23].

Our findings revealed that WNK2 promotes OC progression by upregulating the oncogene POU5F1B. POU5F1B, a homolog of the pluripotency regulator OCT4, contains a complete open reading frame with 96% amino acid sequence

 

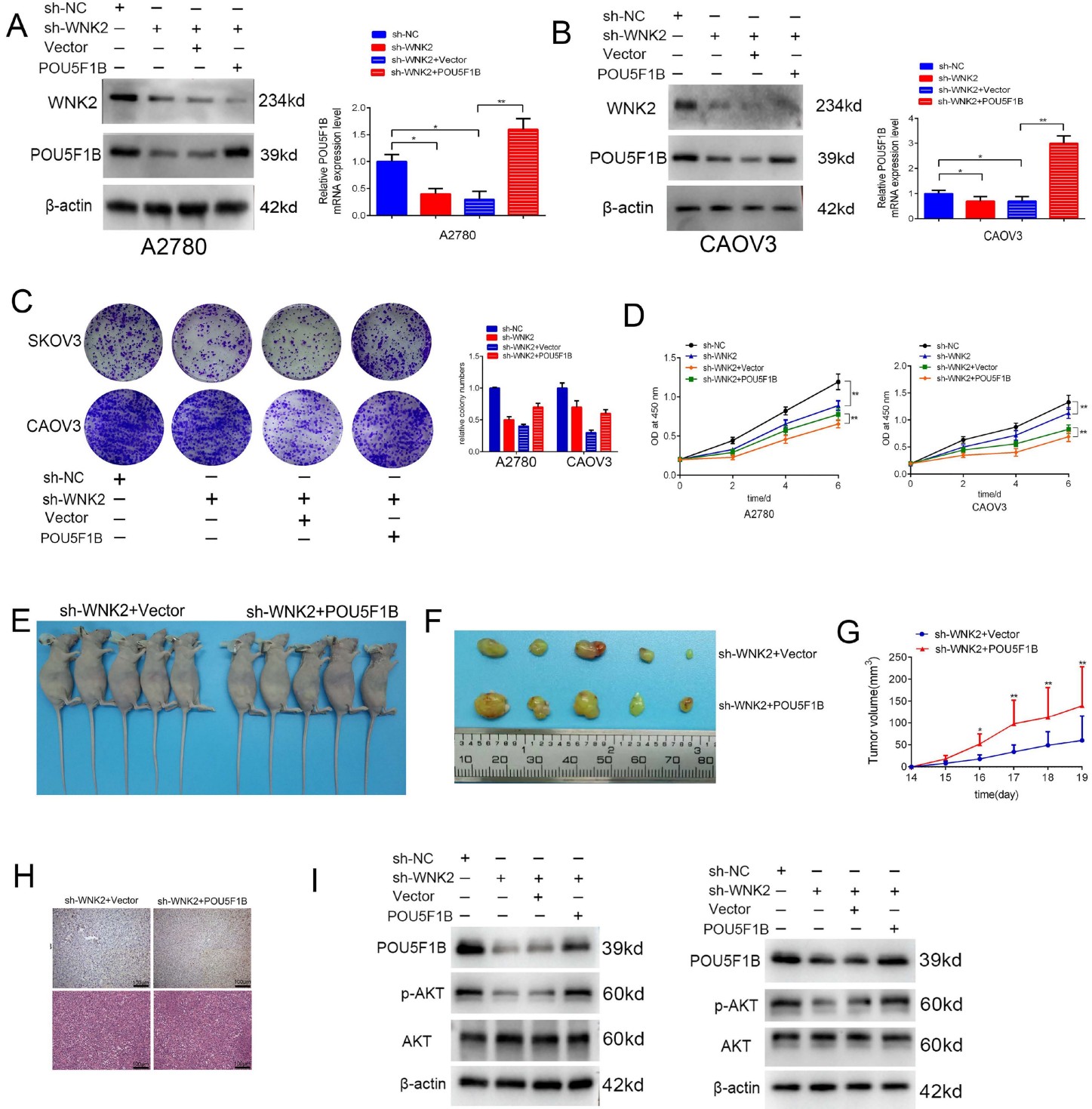

**Fig 6. Oncogenic role of WNK2 partly relies on POU5F1B.** A-B, POU5F1B was successfully overexpressed in the WNK2-knockdown cell lines A2780 and CAOV3. C-D, WNK2 knockdown suppressed the proliferation of OC cells. When POU5F1B was overexpressed, the proliferation inhibition of OC cells was reversed. E-F, POU5F1B promoted the viability of WNK2-knockdown cell in vivo. G, Curves of tumor growth: When visible lumps occurred, tumor volumes were measured every two days. H, Immunohistochemistry staining of POU5F1B in nude mice tumors. WNK2 knockdown inhibited the migration of OC cells. When POU5F1B was overexpressed, the migration inhibition of OC cells was reversed. I, Western blot illustrated the level of p-AKT(473) in OC cells after WNK2 was knocked down and POU5F1B overexpressed.

identity to OCT4 isoform 1. It encodes a closely related protein that interacts with protein kinases and cytoskeletal components, thereby contributing to oncogenic signaling [24–26]. POU5F1B modulates intracellular signaling and trans-acting factors that regulate cell growth and adhesion [27]. Its aberrant expression not only predicts poor clinical outcomes but also drives malignant behavior across multiple cancers [25,28,29]. In this study, we identify the oncogenic function of POU5F1B in OC, demonstrating its overexpression in OC tissues and its ability to promote malignant cellular behaviors. Mechanistically, we establish POU5F1B as a critical downstream effector of WNK2. Our data show that WNK2 promotes OC progression by upregulating POU5F1B, which subsequently activates AKT signaling through phosphorylation at Ser473. Although our findings highlight the pivotal role of WNK2 in OC, its function as a kinase implies that POU5F1B regulation occurs indirectly, likely via an intermediate transcription factor. We propose that WNK2 upregulates POU5F1B expression by activating a transcriptional regulator. Identifying this intermediary factor will be a key focus of future research. Clinically, WNK2 and POU5F1B expression levels in tumor tissues may serve as prognostic biomarkers for OC. Furthermore, our results support targeting the WNK2/POU5F1B/AKT signaling axis as a promising therapeutic approach. We are currently pursuing drug screening efforts to explore this pathway, including evaluating the efficacy of WNK463 – a potent, orally available pan-WNK kinase inhibitor currently used to treat hypertension – as a potential therapeutic agent for OC.

## Supporting information

**S1 File. S1 raw images.**
(PDF)

**S2 File. Raw material.**
(PDF)

## Author contributions

**Conceptualization:** Zhiqing Liang, yanzhou Wang.

**Data curation:** Fengjie Li, Deng Li.

**Funding acquisition:** Zhiqing Liang, yanzhou Wang.

**Investigation:** Fengjie Li.

**Validation:** Yongqin Jia.

**Visualization:** Xiaoli Min, Pangyang Zhang.

**Writing – original draft:** Fengjie Li.

**Writing – review & editing:** Yudi Li, Lanqin Cao.

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
