## [Decision Letter · Decision Letter 0]

14 Oct 2025

Dear Dr. li,

Thank you for submitting your manuscript to PLOS ONE. After careful consideration, we feel that it has merit but does not fully meet PLOS ONE’s publication criteria as it currently stands. Therefore, we invite you to submit a revised version of the manuscript that addresses the points raised during the review process.

We look forward to receiving your revised manuscript.

Kind regards,

Zu Ye, Ph.D.

Academic Editor

PLOS ONE

**Journal Requirements:**

“Key Project of Chongqing Technology Innovation and Application Development Special Project (CSTB2022TIAD-KPX0154).”

3. We note that your Data Availability Statement is currently as follows:

“All relevant data are within the manuscript and its Supporting Information files.”

**Additional Editor Comments:**

Please find attached the detailed comments from the reviewers. We kindly ask you to carefully address each point raised in your revision. When submitting the revised manuscript, please also provide a point-by-point response to the reviewers’ comments, outlining the changes made or explaining your reasoning if any suggestions were not incorporated.

Reviewers' comments:

Reviewer's Responses to Questions

**Comments to the Author**

1. Is the manuscript technically sound, and do the data support the conclusions?

Reviewer #1: Partly

Reviewer #2: Yes

2. Has the statistical analysis been performed appropriately and rigorously?

Reviewer #1: Yes

Reviewer #2: Yes

3. Have the authors made all data underlying the findings in their manuscript fully available?

Reviewer #1: Yes

Reviewer #2: Yes

4. Is the manuscript presented in an intelligible fashion and written in standard English?

Reviewer #1: No

Reviewer #2: Yes

Reviewer #1: In the manuscript entitled “WNK2 facilitates ovarian cancer progression by upregulating POU5F1B”, the authors investigate the role of WNK2 in ovarian cancer progression, suggesting that WNK2 promotes tumor development by upregulating POU5F1B and subsequently activating the AKT signaling pathway.

The study employs mouse models and ovarian cancer cell lines, in which the expression levels of WNK2 and POU5F1B were modulated to validate the proposed model.

Overall, the topic is relevant and potentially contributes to understanding the molecular mechanisms underlying ovarian cancer progression. However, several issues need to be addressed before the manuscript can be considered for publication. Therefore, I recommend a major revision.

Major concerns:

The manuscript requires a thorough review of English grammar and spelling, as several sentences are unclear or contain typographical errors.

Please verify the corresponding author information, which differs between the initial submission details and the main manuscript. The keywords should be revised because also differs from the initial information.

Table 1 is not properly formatted and needs to be restructured for clarity and readability.

In Figure 1, the text refers to liver cancer in subsection 3.1, but this is not mentioned in the figure. Additionally, the figure legend is incorrect, and in Figure 1a, the acronyms (e.g., GBM and others) are not defined.

Figures 3c–d are not mentioned or discussed in the main text. Please review the figure legends and ensure consistency throughout the manuscript.

The quality of the figures should be improved, as they currently appear to be in low resolution and difficult to interpret.

Once these issues are carefully addressed, the manuscript will be in a much stronger position for further consideration.

Reviewer #2: Title

WNK2 facilitates ovarian cancer progression by upregulating POU5F1B

Abstract

Abstract might briefly note clinical implications or sample size in xenograft validation

Introduction

Comprehensive background on ovarian cancer epidemiology and therapeutic landscape.

Strong rationale for exploring WNK2’s dual function

Overreliance on citation stacking; a schematic or model summary would strengthen readability

Methods

Clear description of controls (empty vectors, siRNAs, lentiviral shRNAs).

Ethical approval noted for animal experiments.

Lack of explicit description of RAS inhibition timing or concentration response (Salirasib dosage) may limit interpretability

Results

Data presented chronologically

Data on chapter and in figures/tables equal

There is no interpretation of the results

Could include co-immunoprecipitation or promoter analysis to determine if WNK2 directly regulates POU5F1B transcription

Figures illustrating time-course or dose dependency of AKT activation would enhance clarity

Discussion

Chapter is not started from concise statement summarizing the main findings of the study

Lacks discussion of clinical translation—how WNK2 or POU5F1B could serve as biomarkers or drug targets

Highlighted statement what is offered by the study for future direction

Authors offer to consider of the necessity of prospective study with clearly defined the aims

**Do you want your identity to be public for this peer review?** For information about this choice, including consent withdrawal, please see our Privacy Policy

Reviewer #1: **Yes:** Josiany Carlos de Souza

Reviewer #2: No

---

## [Author Response · Author response to Decision Letter 1]

21 Nov 2025

We extend our deepest appreciation to the reviewers and the editors for their time and insightful comments, which are immensely helpful for improving our manuscript.

Response to editor.

1.We have adopted your suggestion to deposit our laboratory protocols in protocols.io. (DOI dx.doi.org/10.17504/protocols.io.q26g7nn81lwz/v1)

2.We have formatted the article according to the guidelines.

3.We have stated the role of the financial disclosure:“Key Project of Chongqing Technology Innovation and Application Development Special Project (CSTB2022TIAD-KPX0154)” in the study, and included this amended role of Funder statement in the cover letter.

4.The raw data and original uncropped and unadjusted images have been uploaded to the supplementary materials.

5.We have deleted the ethics statement from any other section besides Materials and Methods.

6.In cover letter, we note the blot/gel image data are provided in Supporting Information.

7.We have uploaded the figure files to the Preflight Analysis and Conversion Engine (PACE) digital diagnostic tool, to ensure that figures meet PLOS requirements.

Response to Reviewer 1

1.The manuscript has underwent professional English language editing by Genesis Technology Communication, Co. Ltd.

2.We have properly formatted Table 1 for easier clarity and readability.

3.We have corrected the text in subsection 3.1 and the legend of Fig.1. We also defined the acronyms (e.g., GBM and others) in the manuscript.

4.We have added the description of Figures 3c–d in the main text.

5.We have improved the layout of the graphics for better interpretation.

Response to Reviewer 2

1.We have added the clinical implications or xenograft sample size in the abstract.

2.We have carefully structured the introduction to be more logical.

3.We appreciate your suggestion very much. The AKT kinase is a cornerstone of oncogenic signaling, whose activation drives cancer progression through multiple mechanisms. The finding that POU5F1B can attenuate WNK2-induced AKT activation underscores its pivotal role as a central modulator of this pathway. We found that WNK2 promotes the transcription of POU5F1B, suggesting that it likely activates a specific transcription factor responsible for POU5F1B expression. Because WNK2 is a protein kinase, not a transcription factor. We design phosphorylation sequencing on WNK2, followed by co-immunoprecipitation (Co-IP) of POU5F1B and mass spectrometry. The ultimate goal is to identify and validate the interacting proteins between them. This set of experiments is being planned as the foundation for our next-step work. The work plan for this part is outlined in the Discussion section. By the way, RAS is a fundamental family of small GTPase proteins that act as crucial molecular switches, Given that RAS is not a transcription factor too, and its regulation of POU5F1B is indirect, we have decided to remove this set of experiments from the manuscript to maintain a focused narrative on direct regulatory mechanisms.

4.We appreciate your suggestion, to demonstrate that POU5F1B impacts the activation of AKT by WNK2, we assessed AKT activity following WNK2 knockdown and POU5F1B overexpression. Cells were transfected with plasmids for 48 hours to achieve successful gene overexpression. This is consistent with conditions used in published studies investigating gene's function. While a systematic dose- and time-response analysis is undoubtedly valuable, However, we found it challenging to achieve the ideal dose-/time-dependent effects under our experimental conditions.

5.We have incorporated discussions on the potential of WNK2 and POU5F1B as biomarkers and therapeutic targets in our manuscript.

---

## [Decision Letter · Decision Letter 1]

11 Dec 2025

WNK2 facilitates ovarian cancer progression by upregulating POU5F1B

PONE-D-25-45829R1

Dear Dr. Wang,

We’re pleased to inform you that your manuscript has been judged scientifically suitable for publication and will be formally accepted for publication once it meets all outstanding technical requirements.

Kind regards,

Zu Ye, Ph.D.

Academic Editor

PLOS One

Additional Editor Comments (optional):

Reviewers' comments:

Reviewer's Responses to Questions

**Comments to the Author**

Reviewer #1: All comments have been addressed

Reviewer #2: All comments have been addressed

2. Is the manuscript technically sound, and do the data support the conclusions?

Reviewer #1: Yes

Reviewer #2: Yes

3. Has the statistical analysis been performed appropriately and rigorously?

Reviewer #1: Yes

Reviewer #2: Yes

4. Have the authors made all data underlying the findings in their manuscript fully available?

Reviewer #1: Yes

Reviewer #2: Yes

5. Is the manuscript presented in an intelligible fashion and written in standard English?

Reviewer #1: Yes

Reviewer #2: Yes

Reviewer #1: (No Response)

Reviewer #2: After reviewing the revised version of the manuscript, I acknowledge that the authors have made several improvements compared with the previous submission. They addressed a number of editorial and reviewer comments, including deposition of protocols, addition of experimental details, and removal of the RAS-related experiments. The manuscript is now clearer in several sections, and the figures are more consistent with the described results

**Do you want your identity to be public for this peer review?** For information about this choice, including consent withdrawal, please see our Privacy Policy

Reviewer #1: No

Reviewer #2: **Yes:** Raikhan Bolatbekova

---

## [Editor Report · Acceptance letter]

PONE-D-25-45829R1

PLOS One

Dear Dr. Wang,

I'm pleased to inform you that your manuscript has been deemed suitable for publication in PLOS One. Congratulations! Your manuscript is now being handed over to our production team.

Kind regards,

on behalf of

Prof. Zu Ye

Academic Editor

PLOS One